# The Prebiotic Kitchen: A Guide to Composing Prebiotic Soup Recipes to Test Origins of Life Hypotheses

**DOI:** 10.3390/life11111221

**Published:** 2021-11-11

**Authors:** Lena Vincent, Stephanie Colón-Santos, H. James Cleaves, David A. Baum, Sarah E. Maurer

**Affiliations:** 1Wisconsin Institute for Discovery, University of Wisconsin-Madison, Madison, WI 53715, USA; lvincent3@wisc.edu (L.V.); colonsantos@wisc.edu (S.C.-S.); 2Earth and Planets Laboratory, The Carnegie Institution for Science, Washington, DC 20015, USA; henderson.cleaves@gmail.com; 3Earth-Life Science Institute, Tokyo Institute of Technology, Ookayama, Meguro-ku, Tokyo 152-8550, Japan; 4Blue Marble Space Institute for Science, Seattle, WA 97154, USA; 5Department of Botany, University of Wisconsin-Madison, Madison, WI 53705, USA; 6Department of Chemistry and Biochemistry, Central Connecticut State University, New Britain, CT 06050, USA

**Keywords:** prebiotic chemistry, prebiotic synthesis, prebiotic soup, prebiotic mixture, origin of life

## Abstract

“Prebiotic soup” often features in discussions of origins of life research, both as a theoretical concept when discussing abiological pathways to modern biochemical building blocks and, more recently, as a feedstock in prebiotic chemistry experiments focused on discovering emergent, systems-level processes such as polymerization, encapsulation, and evolution. However, until now, little systematic analysis has gone into the design of well-justified prebiotic mixtures, which are needed to facilitate experimental replicability and comparison among researchers. This paper explores principles that should be considered in choosing chemical mixtures for prebiotic chemistry experiments by reviewing the natural environmental conditions that might have created such mixtures and then suggests reasonable guidelines for designing recipes. We discuss both “assembled” mixtures, which are made by mixing reagent grade chemicals, and “synthesized” mixtures, which are generated directly from diversity-generating primary prebiotic syntheses. We discuss different practical concerns including how to navigate the tremendous uncertainty in the chemistry of the early Earth and how to balance the desire for using prebiotically realistic mixtures with experimental tractability and replicability. Examples of two assembled mixtures, one based on materials likely delivered by carbonaceous meteorites and one based on spark discharge synthesis, are presented to illustrate these challenges. We explore alternative procedures for making synthesized mixtures using recursive chemical reaction systems whose outputs attempt to mimic atmospheric and geochemical synthesis. Other experimental conditions such as pH and ionic strength are also considered. We argue that developing a handful of standardized prebiotic recipes may facilitate coordination among researchers and enable the identification of the most promising mechanisms by which complex prebiotic mixtures were “tamed” during the origin of life to give rise to key living processes such as self-propagation, information processing, and adaptive evolution. We end by advocating for the development of a public prebiotic chemistry database containing experimental methods (including soup recipes), results, and analytical pipelines for analyzing complex prebiotic mixtures.

## 1. Introduction

Since the pioneering research by Miller and Urey in the 1950s [1,2], it has gradually become accepted that abiotic synthesis in the atmosphere, hydrosphere, and lithosphere, combined with exogenous inputs from space, likely provided prebiotic Earth with a diverse inventory of organic compounds [3,4,5]. These considerations imply that bodies of water on the prebiotic Earth were imbued with a chemically diverse organic content, a so-called “prebiotic soup” [6]. The chemical composition of this potentially multiply sourced mixture is uncertain beyond the inference that it was chemically diverse and likely included many of the important chemicals involved in cellular metabolism and genetics, albeit at perhaps very low steady-state concentrations.

While it is well documented that many chemicals involved in biochemistry can be synthesized abiotically [7], the biggest outstanding problem in understanding the origins of life is how the components of prebiotic soup came to be organized in systems capable of emergent processes such as growth, self-propagation, information processing, and adaptive evolution [8,9,10,11]. Given that prebiotic soups may have been composed of millions of distinct compounds, each at a low concentration [12,13], another mystery is how processes winnowed this molecular diversity down to the few compounds it used by biology today, which are a tiny subset of the many compounds that would have arisen from abiotic processes. Consequently, it is important to understand how complex mixtures of dilute organic molecules generated by environmental processes could have been “tamed” to give rise to the less diverse but more organized chemistry of metabolism [14,15,16,17].

Understanding the taming of chemical complexity and the emergence of key life processes likely requires “bottom-up” experiments [18], which entail studying how model prebiotic mixtures converge towards life in terms of the spectrum of chemicals formed, their relative abundances, or their overall dynamical behavior. The starting point for such experiments should be mixtures of chemicals (“soups”) that could plausibly have been present on early Earth. Using such experimental inputs, many questions can be addressed. For example, one could ask how prebiotic mixtures are modified upon interacting with minerals or upon exposure to environmental fluctuations such as wet–dry cycling. Experiments might be conducted in materially closed systems or may simulate the flow of soup through a primordial microenvironment, for example, by periodically replenishing reagents [19,20,21] or using a continuous flow reactor [22,23]. Whether one is searching for the emergence of particular chemicals (e.g., nucleotides, amphiphiles, polymers) [1,24,25,26], for autocatalysis, or for other life-like dynamical properties [20,27], experimental results are likely be sensitive to the chemicals used in the input solutions.

How can appropriate soups be designed in the face of the tremendous uncertainty regarding the prevailing chemistry in any given locale on early Earth? How can the desire for using prebiotically realistic soups best balance experimental tractability and replicability? These important practical questions have not yet been adequately discussed. The aim of this paper is to explore the principles and practicalities of designing prebiotic soups for bottom-up origins of life research. Two complementary approaches for making prebiotic soups are considered: assembling them and synthesizing them. “Assembled soups” are made by combining laboratory-grade chemicals, while synthesized soups are generated via recursive, diversity-generating chemical reactions starting from a small number of low molecular mass input chemicals.

After proposing some general principles guiding the design of experimental prebiotic soups, we discuss the challenges that arise from the uncertainty regarding early Earth conditions and the considerable temporal and spatial variation of geochemical conditions that likely existed. We suggest that since the successful bottom-up origins of life research program is only feasible if abiogenesis is a reasonably robust phenomenon, meaning that it does not require very specific, cosmically rare conditions, the desire to generate a perfect simulacrum of prebiotic chemistry should not prevent attempts to generate reasonable approximations that bracket some of the uncertainty. To illustrate these principles, we provide a handful of prebiotic soup recipes and then argue for community coordination, perhaps including the generation of a shared repository of soups and recipes so as to add rigor and repeatability to the study of complex prebiotic chemistry.

## 2. General Principles and Challenges for Designing Experimental Prebiotic Soups

Origins of life research programs can generally be characterized based on whether they aim to address the historical question of the emergence of biochemistry on Earth or the ahistorical question of how life as a general phenomenon arises [28]. This is a spectrum rather than a discrete distinction. It may never be clear if we have solved the historical problem of how life actually arose on Earth, so historical research is to some degree concerned with identifying conditions under which life *could* evolve. Conversely, a research program can only be said to solve the ahistorical problem if it uses mixtures and conditions that might realistically occur in at least one natural environment, somewhere in the Universe. Engineering life in an artificial lab setting would not explain how life could emerge spontaneously. Furthermore, whether research is at the more historical or ahistorical ends of the spectrum, there will be a trade-off between inferred geological realism and expediency. In practice, all experiments sacrifice some degree of realism. For example, the original Miller–Urey experiment [1] was guided by ideas regarding the reducing atmosphere of the early Earth [6,29], but did not attempt to simulate the effects of all possible energy sources (e.g., ultraviolet radiation) or the presence of minerals.

Even restricting our attention to prebiotic soups in aqueous solution, there is surely no single correct solution to use. Even if there is some desire to focus on historical origins, soups that are made to be realistic facsimiles of Earth’s prebiotic oceans might be different from those made to mimic lakes or other microenvironments, let alone water bodies elsewhere in the Universe. Nonetheless, there are likely to be more and less reasonable soups for any given targeted environment, posing the question of how these mixtures should be designed.

Two main strategies for generating prebiotic soups suggest themselves. The first is an assembled soup, made by dissolving reagent-grade commercial chemicals and mixing them in proportions mimicking environments of interest. The second is a synthesized soup, made by standardizing a prebiotically plausible mixture of small organic chemicals and then allowing them to react under specified conditions. These two strategies have complementary strengths and weaknesses (Table 1).

Assembled soups have the advantage that they can be produced without complex chemical reactors and should be similar between experimentalists (at least insofar as the same chemical sources are used). The fact that an assembled soup’s composition is largely known (although not entirely due to impurities and reactions among the soup’s components) is potentially useful for replication and comparison among conditions. Additionally, selective subtraction or isotopic labeling of some components might allow elucidation of key chemical mechanisms. On the negative side, the total number of distinct compounds in an assembled soup will likely be lower, and the relative concentrations of individual components higher, than strict realism might demand.

Synthesized soups have the advantage of starting with few ingredients at relatively high concentrations yet yielding highly complex mixtures through recursive chemical processes, including many compounds at extremely low concentrations. As a result, synthesized soups might more accurately mimic prebiotic concentration profiles. On the other hand, synthesized soups starting from gases may be more difficult to produce under laboratory conditions in the absence of specialized equipment and safety protocols, a problem that can be avoided using liquid synthesis methods. Furthermore, synthesis experiments are likely to be sensitive to the choice of starting chemicals, the reaction conditions, and the presence of minerals or trace ingredients. In all cases, the design of a soup should be guided by the best available predicted geochemical/environmental conditions for the desired location.

## 3. Prebiotic Sources of Organics and Challenges for Soup Design

### 3.1. Terrestrial Sources of Organics

Dissolved organics on primitive Earth could have derived from a combination of exogenous sources (comets, meteorites, and IDPs) and endogenous sources [30], which includes both atmospheric and hydrothermal synthesis. One of the challenges with understanding the chemical inputs from these sources is that each would have been highly dependent on imperfectly known aspects of the Hadean environment.

Geothermal energy may have plausibly driven prebiotic synthesis in some contexts, since mineral surfaces and high pressure and temperature provide conditions favorable to the generation of organic compounds [31]. At high pressure and temperature, high concentrations of CO_2_ and H_2_ in the presence of metal catalysts (e.g., iron-sulfur clusters; [32,33]) can produce organic compounds, including membrane forming amphiphiles like fatty alcohols and fatty acids [34,35]. These Fischer-Tropsch-Type reactions, may be possible in hydrothermal systems, although only small amounts of hydrocarbons and fatty acids have been detected in modern environmental samples [36]. However, mid-ocean ridge vent fluid can be rich in reduced gases like hydrogen [37], which, in a prebiotic context, could have reacted with prebiotic chemicals generated from other mechanisms to drive early metabolism [17,38].

Atmospheric synthesis likely provided an additional source of endogenous organics. Miller (1953) showed that when an electric discharge is passed through a reducing gas mixture, similar to that envisioned by Oparin in the 1920s [29,39], organics were readily generated. It has been estimated that ~4 Ga, between ~4 × 10^8^ and 2 × 10^11^ kg yr^−1^ of organics were produced from atmospheric reactions, depending on the atmospheric oxidation state [30,40], which would correspond to a material influx to the oceans of between 0.8 and 390 mg m^−2^ yr^−1^. These high production rates may give a false sense of abundance: if these organics were entirely composed of glycine, and this were all dissolved in oceans of the modern volume (~1.35 × 10^21^ L), a year’s production would generate a glycine concentration in the picomolar to low nanomolar range. Assuming no loss, even after a million years, concentrations might still be too low for many types of reactions to occur without additional concentrating mechanisms.

The types of products made via atmospheric synthesis are sensitive to the types and abundances of gases present and the energy sources used (e.g., UV, spark-discharge, shock waves) [41,42]. There is no consensus regarding the oxidation state of Earth’s early atmosphere during the period from ~4.3 Ga to 3.5 Ga which most scientists consider to be the window for the origin of life. While it was likely neutral (composed mostly of N_2_ and CO_2_), there may have been sporadic reducing periods after volcanic activity or meteorite impacts [43,44], generating conditions thought to be more conducive to atmospheric organic synthesis both in terms of compound diversity and abundance [41,45,46]. Furthermore, since mineral surfaces and dissolved inorganic species can greatly alter prebiotic chemistry [47,48], and since the types of inorganic species would have been highly dependent on environmental context, there are reasons to expect considerable variation across microenvironments such as deep sea hydrothermal vents, tidal pools, and rain-fed lakes and ponds.

In addition to spatial heterogeneity and uncertainty as to the chemical conditions at any one time and place, there was likely considerable temporal variation in chemical conditions. As well as periodic extraterrestrial impacts, mentioned above, heat flow from Earth’s interior has been decreasing over time [49] and photon flux from the Sun has varied both in terms of flux and spectral intensity [50]. There is also the question of when Earth’s geodynamo became active, which would have affected the rate of atmospheric hydrogen loss on early Earth and the efficacy of solar energy-mediated atmospheric and surface synthesis [51,52].

Nonetheless, despite all this uncertainty, the Miller–Urey experiment and subsequent studies revealed that a handful of small reactive organic compounds can form readily, including formaldehyde and hydrogen cyanide (HCN). This is significant because the autocatalytic formose reaction [53], commonly invoked as an abiotic source of sugars, can be initiated by the photochemical formation of formaldehyde from water and carbon dioxide [54,55]. Given the variability in the products of endogenous prebiotic syntheses and its dependence on starting conditions, bottom-up experiments aiming to mimic endogenous sources should perhaps prioritize the inclusion of diverse organics and key reactive compounds (e.g., HCN) rather than attempt to perfectly replicate the actual synthesis mechanisms.

### 3.2. Exogenous Delivery of Organics

The synthesis of organics in space and their delivery to Earth via interplanetary dust particles (IDPs), meteorites, and comets is another potentially important source of organics [40,56]. Such materials would have delivered organic molecules during and after the accretion of the planet. Many scientists believe that the influx of extraterrestrial materials to Earth decreased exponentially over time, although it may have been punctuated by periodic heavy bombardment [57,58]. It is estimated that 6 × 10^7^ kg yr^−1^ of organic material were delivered to Earth around 4 Ga, corresponding to a surface-averaged flux of ~0.1 g m^−2^ yr^−1^, which is significantly less than that estimated from endogenous atmospheric reactions [30]. It should also be noted that most organics are not indefinitely stable in aqueous environments, especially at high temperatures and pH values, and thus steady state concentrations may have been somewhat lower [59].

Though likely not the major exogenous source of organics to early Earth, carbonaceous chondrite meteorites have been the focus of considerable study due to their diverse organic contents. These meteorites make up only approximately 4% of all meteorites in collections but contain significant amounts of organic materials that sometimes give evidence for thermal, aqueous, and radiation alteration over their long tenure in space [60]. In some cases, this processing seems to have generated a large diversity of organic compounds [61], including enantiomeric excesses in certain compound classes [62]. The compounds formed and their abundances may depend on the extent of processing experienced by the specific meteorite [26], which indicates that trying to replicate a specific meteoritic composition exactly may be unimportant; it may be more important to aim for a diversity of molecules with concentrations similar to those measured in a typical carbonaceous meteorite. There is significant overlap of abundance patterns between the composition of carbonaceous meteorites and laboratory spark-discharge experiments, which points to there being some similarity in synthetic mechanisms (e.g., Wolman et al., 1972 [63]).

Undoubtedly, the best-characterized carbonaceous chondrite is the Murchison meteorite, whose organic components have been extensively catalogued. Amino, hydroxy, and carboxylic acids are among some of the important biologically relevant components [64], though it should be borne in mind that untargeted analyses suggest there may be several million relatively low molecular weight compounds present [12], and thus the compounds of biological relevance are only a small fraction of the non-biological suite. Nevertheless, the Murchison meteorite is used in the present paper to provide an example of a “meteoritic soup” recipe for origin-of-life studies, for which chemical details and assembly instructions are presented in the Appendix A.

## 4. How to Make Prebiotic Soup

As summarized in Figure 1, we outline and discuss considerations for the design of both assembled and synthetic soups, and then provide examples of recipes and procedures to illustrate the overall approach. The technical details accompanying these examples can be found in the SI. These examples can be used directly, be modified to suit a different type of experiment, or inspire others to search the literature and generate a novel recipe. When using either a synthetic or assembled mixture for experimental purposes, the “bottom-up prebiotic experiments” (bottom of Figure 1) would directly impact the design rationale for mixture preparation, as elaborated below.

### 4.1. Assembled Prebiotic Soup

To assemble a soup, commercially available chemicals are mixed according to a pre-defined recipe. One of the major advantages of assembled soups is that downstream analysis is much easier with a smaller number of compounds than in a synthesized soup. Additionally, it is relatively easy to control chirality by adding specific enantiomers to an assembled soup, something that cannot be done with synthesized soups, or to track chemical transformations by including isotopically labeled compounds. However, assembled soups also have important disadvantages. Extracts from the Murchison meteorite have been found to contain tens of thousands of distinct CHONSP-containing molecular formulas, with a likely even greater number of distinct molecular structures [12], which suggests that mixing even a few hundred compounds in solution would not mimic the true molecular diversity of prebiotic organics. However, it is also possible that all of the main chemical reactions that occur among the many chemicals present in a prebiotic mixture would occur in a less diverse mixture. For example, even though many *α*-amino acids are generated by prebiotic synthesis, the reactivity of the carboxylate group may be similar for the entire compound class. This implies that it may be more important for assembled soups to sample major organic functional groups and chemical reactivities than to include the full diversity of chemical compounds.

Based on these considerations, assembled soups may be preferable if an experimental program (1) requires having control over the composition of the soup, including the addition of chiral compounds, (2) does not require a high diversity of products, (3) is focused on particular types of chemistry or compound class rather than on the full spectrum of chemical possibilities. In the following sub-sections, we discuss how to select compounds, set their concentrations, decide on chirality, and approach other aspects of soup composition.

#### 4.1.1. Selection of Compounds

There are several factors that should be weighed when deciding which compounds to add to a prebiotic soup recipe, including the hypothesized source(s) of prebiotic organics and the processes or features arising from prebiotic soup that are under study. A concern with synthesized soups is that the chemicals that are detected and abundant as products of abiotic processes will be those that are most stable and, thus, less reactive. In cases where an influx of chemicals is expected in a new environment (for example, being washed into a hot spring) mixtures of the more stable compounds are perhaps justified. However, when modeling environments in which chemicals are produced in situ, short-lived reactive species (e.g., radicals) may have had an important role [65]. Such considerations recently prompted researchers to use a combination of assembly and synthesis, where specific laboratory grade chemicals were combined with a synthetized prebiotic soup to characterize potential interactions [66].

The compounds that are ultimately included in an assembled soup will also depend on the recipe’s compatibility with selected environmental conditions. Examples of issues that can arise include insolubility at a given pH or in the presence of inorganic species. Attention should also be given to the process or phenomenon that is being studied in an experimental program. For instance, prebiotic soups designed for studies of autocatalytic behavior may prioritize chemical diversity, while research into specific chemical pathways or reaction types might favor higher concentrations of a few focal compounds. To illustrate some of the decisions and tradeoffs that need to be made in designing assembled prebiotic soups, two examples are described: a meteorite soup and a spark-discharge soup. We use these two examples for the sake of discussion throughout this section.

The proposed meteorite recipe is based on data from the Murchison meteorite (Appendix A). Any chemical compound detected with a concentration greater than 10 nmol g^−1^ meteorite that was commercially available/affordable was included [61,67,68,69,70,71,72,73,74,75,76,77,78]. We used a systematic approach based on a threshold concentration to make decisions about which compounds to add. This narrowed the list down to a more manageable number. Importantly, we checked that this subset of chemicals included compounds representative of each major compound class (e.g., amino acids, alkanoic acids, etc.) detected in the meteorite.

A different approach was used to design the spark-discharge soup (Appendix A). The general strategy was to use data on the chemical outputs from multiple sources to identify compounds present in multiple experiments carried out under different conditions [2,41,79,80,81]. To compare concentrations across separate studies, which sometimes characterized different compound types (amino acids, organic acids, nitriles, nucleobases, etc.), we used a benchmark species (glycine) that appeared consistently. We did not include all classes of compounds that have been reported in spark-discharge experiments [82,83], focusing instead on those with potential roles in metabolism-like processes. Thus, for example, we omitted hydrocarbons and fatty acids, although these could be added readily in the future. Likewise, we opted not to include every possible chemical detected within a compound class. For instance, we only included cysteine rather than other sulfur-containing amino acids and excluded decomposition products and intermediates that were sometimes detected. Such simplification is guided by the particular chemical questions being asked. For example, if one were specifically interested in sulfur chemistry and its role in the emergence of life processes, it would make sense to assemble a soup with a wider variety of sulfur-containing compounds, while perhaps reducing the number of nitrogen-containing compounds. Details on the approach can be found in the SI.

There are obvious pros and cons to each recipe. The meteoritic soup is based on mixtures that were highly likely on early Earth, and not biased by modern biology (except to the extent that the compounds detected to date may be biased by the interests of analytical chemists). However, adding 80+ organics to a solution is difficult and the resulting solution may include insoluble components. The spark-discharge soup has fewer components which are all water soluble, making it easier to prepare, work with, and analyze. However, the selected compounds are more biased toward extant life and based on less rigorous, more arbitrary criteria for inclusion.

#### 4.1.2. Setting Concentrations

Realistic concentrations of organics in primitive aqueous planetary environments are difficult to estimate. Even if all the organics produced over half a billion years were to accumulate in oceans of the modern volume (how the volume of the oceans has changed over Earth history is also debated; [84]) the solution would still be very dilute. Nonetheless, concentrations could be higher in local environments due to concentration mechanisms such as evaporation [85], eutectic freezing [86,87], or the dehydration of aerosols [88]. Regardless of absolute concentrations, it seems wise to include reactants in molar ratios similar to those detected in prebiotic simulations or natural samples.

To illustrate different potential approaches for setting concentrations, we refer back to our two working examples. For the meteorite soup, we generated a 1 L solution with 100 g of organic material. Each selected compound was taken as representative of a particular compound class and the amount added was selected to maintain the relative concentrations of compound classes seen in Murchison. This is a very concentrated soup, but it can be diluted arbitrarily to make a working solution. Using the provided spreadsheet, compounds can be added by changing the “include” value to “yes”, which will shift all the gram amounts within that compound class without changing the combined amount of that compound class. Additionally, the compound classes can be modified to represent other meteorite compositions. This spreadsheet can also serve as a starting point for assembling other kinds of soups.

For the spark-discharge soup, the absolute concentrations of individual compounds were increased compared to those reported in spark-discharge synthesis experiments to (1) help with downstream detection of new products (by raising their concentration above the limit of detection of analytical instruments), and (2) facilitate the preparation of the soup by avoiding weights and volumes that are too small to be handled and measured reliably. To set concentrations, we first fixed the concentration of glycine and used approximate molar ratio data from the references listed in Appendix A to adjust the relative concentration values of the remaining compounds. To select concentrations, we assigned ranges of concentrations reported in the original literature to specific concentration values, in multiples of 0.08 mM (see SI for exact method and concentration conversion multipliers), the lowest concentration value in our example recipe. This was an arbitrary decision made to simplify the task of setting concentrations for each compound in the assembled soup recipe.

#### 4.1.3. Chirality

Most prebiotic reactions conducted to date have been found to produce roughly racemic product suites, which contrasts with the enantiomeric excess (ee) seen in some meteoritic organics [89]. It is unclear whether the ee seen in meteorites arose from the enhancement of small initial fluctuations by autocatalytic reactions [90] or by differences in stability/reactivity [91]. For many purposes, for example to see if ee enrichment emerges spontaneously, it may be preferable to assemble soups from racemic components. Thus, the meteorite soup uses racemic ingredients. Although it should be noted that supposedly racemic compounds from a commercial source almost always show a significant ee [92], it would be possible to measure this ee before and after an experiment to evaluate chiral enrichment. In case of the spark discharge soup, biological entantiomers were chosen, for example L-amino acids, for the simple reason that they are more readily available and less expensive. As a result, this soup, as designed, would not be suited for applications that focused on the origins of ee.

### 4.2. Synthesizing Prebiotic Soup

There are many potential ways to make a synthesized prebiotic soup, but a key concern is defining reagents and conditions that will be minimally sensitive to experimental variation. The first decision is whether to start synthesis from atmospheric gases or to conduct experiments in the liquid phase, starting with the water-soluble, reactive products of gas-phase synthesis. While the former may yield a more authentic soup, gas-phase synthesis methods are significantly more challenging and may not be practical for many research groups. We review each of these approaches below, focusing on gas-phase synthesis with spark discharges and “polymerization” reactions using small, reactive organic inputs (formaldehyde, formamide, and HCN). Generic recipes for each of these can be found in the SI.

#### 4.2.1. Gas Phase Synthesis

The most famous gas-phase prebiotic synthesis strategy uses spark discharges [1], which is known to generate an array of small organics, including both proteinogenic and non-proteinogenic amino acids (see Appendix A). Although a video protocol for conducting spark discharge experiments has been published [93], the procedure is somewhat complex, hazardous if not conducted with some precision, and difficult to conduct in a high-throughput fashion. Some lack of reproducibility has also been reported, which may be due to nuances in experimental design [94]. Nonetheless, if one strives for the most realistic primordial soup, direct, gas-phase synthesis might be the most appropriate strategy.

When using spark discharges, the material of the electrodes can have an effect, with tungsten oxide being the historically preferred option (mainly due to the coefficient of thermal expansion of tungsten oxide which allows it to be easily fused with laboratory glass). Likewise, the temperature to which the water is heated is not necessarily constant across experiments. Furthermore, reaction time likely has an effect. The relative volumes of gas phase to aqueous phase reservoirs likely also matter [13,93,94], a small amount of gas reacted over a large volume of water likely gives a different result than a large volume of gas reacted over a small volume of water. We therefore recommend that if spark discharges are used to generate soup, these variables be rigorously standardized to maximize repeatability.

The product composition of gas-phase synthesis is known to depend sensitively on the gas mixture [95]. Due to the current debate on predominant atmospheric conditions on early Earth, the gas composition applied in spark-discharge experiments can range from reducing (i.e., CH_4_, NH_3_, and NH_3_) to neutral (i.e., CO_2_ and N_2_) [13,81]. Atmospheric synthesis is subject to its own complexities which depend on input species, energy fluxes, and reaction and rainout rates, thus it seems unlikely a simple standardized set of compositions can be defined with regard to these variables.

#### 4.2.2. Liquid Phase Synthesis

Given the practical challenges of gas-phase processes, the alternative is to use liquid-phase syntheses, which start with small, reactive organic species that are known to be produced in abundance in many gas-phase contexts. Perhaps the simplest mechanism for generating a large molecular library is through HCN polymerization [96], which is of particular interest due to the importance of HCN in the formation of nucleobases [97] and amino acids via the Strecker synthesis [75]. However, HCN polymerization should only be attempted by experienced chemists since there is a risk of releasing the poisonous cyanide gas into the laboratory.

Given the hazards of HCN, there has been an increasing interest in the chemistry of formamide (see SI for a generic recipe), which can act as a solvent as well as a reactant for the synthesis of a variety of biochemical compounds and is much easier to handle [98]. Formamide is the first hydrolysis product of HCN and is a ubiquitous molecule in the Universe [99]. Studies have demonstrated that heating formamide in the presence of different catalysts of terrestrial and meteoritic origin yields complex combinations of nucleobases, amino acids, sugars, amino sugars, and condensing agents [81,100]. Formamide reactions also result in selective synthesis of certain nucleobases and nucleosides when mineral surfaces are added to the reaction [101,102].

Time of reaction is also a factor since many radicals and unstable intermediates may be present while the reaction is actively occurring but may be absent as the starting chemicals become depleted and the soup converges on an equilibrium composition. As a result, factors that alter reaction rates, such as pH, temperature, and the concentration of the starting materials need to be controlled. For example, when the formose reaction is carried out at high temperature, the solution quickly turns into a complex intractable mixture, due to competing mechanisms such as the Maillard reaction, degradation, and uncontrolled polymerization of the carbohydrates, but this can be avoided by performing the reaction for short periods of time and in moderate temperatures [103].

The outcomes of liquid-phase synthesis using HCN or formamide depend on concentration, pH, and temperature [59,104,105]. In the case of formamide condensation, typical conditions involve heating pure formamide at 160 °C (the boiling point of formamide is 210 °C) in the presence of inorganic catalysts or UV irradiation. However, the synthesis of nucleobases from formamide has been demonstrated at lower temperatures (i.e., 50 °C) using longer reaction times and recursive addition of formamide [19]. An alternative protocol, developed by Ricardo et al. (2004) [106], uses relatively dilute inputs, borate minerals, alkaline pH (8–11), and a 2-month incubation to promote ribose formation and inhibit the generation of tar.

### 4.3. Inorganic Components

Even after one has decided on a standard way to assemble or synthesize the organic components of soup, inorganic components need to be considered. In large part, choice of experimental variables will be governed by the particular microenvironment researchers aim to mimic, which might range from shallow surface environments such as subaerial hot springs [107,108] to deep ocean environments such as hydrothermal chimneys [109]. Major factors which distinguish these various environments include dissolved inorganic and ionic components, as well as minerals that can affect pH.

Inorganic species, whether dissolved or in the solid phase, can influence synthesized soups (e.g., Surman et al., 2019) [21]. Thus, it will generally be more realistic to conduct syntheses in solutions that already contain relevant inorganic components. All surface waters on Earth contain significant amounts of dissolved inorganic salts. Indeed, many prebiotic chemistry exploration experiments have considered the effects of dissolved salts [110,111]. Seawater-like ionic solutions, however, do introduce experimental difficulties, especially for mass spectrometry and NMR investigations. Instead of synthesizing soups in the presence of inorganic species, it is also possible to add salts and other inorganic solutes after soup assembly/synthesis. If mimicking an ocean environment, an inexpensive and simple starting point for the aqueous phase could simply be to add modern sea salt, which is available as a commercial product (although modern seawater contains significant amounts of sulfate, which may not have been the case in the primitive oceans). However, the composition of seawater has likely changed markedly over time, and the early oceans may have had up to twice the salinity of the modern oceans (35–70 g L^−1^) [112]. If a prebiotic simulation attempts to mimic prebiotic pond or river water, ionic concentrations might be considerably lower (less than 1 mM for Na^+^, Cl^−^, Ca^+^, Mg^+^, K^+^) [113].

The inorganic components of the soup that may have had the greatest role in nascent biochemistry include polyatomic ions containing nitrogen, phosphorus, sulfur [114,115]. Early sea water may also have contained more carbonate, due to higher atmospheric CO_2_ levels [112,116], more sulfur in the −2 oxidation state (compared to the +6 state) [117] and significant amounts of Fe^2+^ [118]. These additions may be very challenging to use when working with a prebiotic soup in the lab as they are sensitive to oxygen in our modern atmosphere. One possibility is to use a glove box or anaerobic chamber to reproduce the anoxic atmosphere and avoid an undue influence of oxygen on experiments [119].

pH may be one of the most important variables in directing prebiotic synthesis. The pH of natural water varies widely, with extreme values ranging from pH 0 to pH 13 [120,121], although modern rivers average a pH of 7.4 [122], and modern ocean water is near pH 8.1 [122]. Ocean pH has likely varied over time, but was likely lower in the Hadean than today due to higher atmospheric pCO_2_ [116]. When adjusting the pH in an assembled soup, pH can either be left unmanipulated, adjusted to a target using simple acids or bases (e.g., HCl, NaOH), or adjusted and kept within a target range by including buffering components in the soup (e.g., acetate/acetic acid), although buffering raises its own set of experimental issues (e.g., concentration of buffer relative to reactants, common ion effects, etc.).

### 4.4. Storage and Transport

As a practical matter, it may not always be possible for a soup to be consistently assembled or synthesized immediately before each experiment. Thus, thought needs to be given to soup storage. A soup is likely to be most out of thermodynamic equilibrium when first prepared and to react and dissipate this disequilibrium over time. These changes are likely to result in soups that become progressively less able to sustain life-like reactivity over time because life usually entails energy dissipation.

In the case of synthesized soups, the ideal approach would be to conduct syntheses simultaneously with their use in experiments, as might be possible using continuous flow reactors. Failing that, the soup needs to be stored in the most inert form possible, which probably requires rapid freezing at −80 °C or below or, perhaps, freeze-drying, although the chemical consequences of drying and re-hydration need to be considered. The same is true for assembled soups, except that it is also advisable to break the soup into two or more “subsoups” that can be stored separately and mixed together just prior to each experiment. This way, one can keep more reactive subsets of chemicals separate until the time of experimentation, which might allow more out-of-equilibrium chemical reactions to occur. We also recommend dividing up the total soup volume into smaller aliquots to avoid the negative effects of serial freeze–thawing cycles on the integrity of the chemical constituents. It may also be preferable to add unstable and/or temperature sensitive components after sterilization or immediately prior to use, in a manner analogous to microbiological media preparation that add sensitive components such as antibiotics at the last minute.

Regardless of the care taken to store soups, it is important to design experiments such that degradation in storage or during freezing and thawing does not yield misleading results. Minimally, some kind of negative control is needed to be able to detect chemical differences that are due to differences among batches of soup or the same batch of soup thawed at different times. For example, Vincent et al. (2019) conducted a chemical ecosystem selection experiment with an assembled soup in which experimental vials with an accumulated history of transfer were always compared to a control set of vials that were set up with the same soup and at the same time but lacked a history of transfer. Indeed, given the high probability that no two batches of primordial soup will be chemically identical, the careful design of experimental controls is absolutely critical for all prebiotic chemistry experimentation.

## 5. A Shared Infrastructure for Complex Prebiotic Chemistry

If our goal is to explore the space of possible soups, physical conditions, and experimental designs to find those that yield life-like chemical phenomena (e.g., Cleaves, 2013) [123], then a community-wide effort is needed. To efficiently scan the parameter space and identify conditions conducive to the emergence of life processes, there is a need to systematically track which parts of the space have already been investigated, how they have been investigated, and whether those conditions yielded positive (or at least interesting) results. Community coordination could facilitate such work by offering standardized procedures for assembling or synthesizing soups, and by sharing a few commonly used experimental paradigms. In particular, it would be useful to establish an online database for depositing and accessing information about specific recipes, synthesis procedures, experimental conditions, etc.

In addition to facilitating inter-lab coordination and a more productive exploration of different experimental parameters, a valuable consequence of sharing information in a “prebiotic soup database” would be the availability of data related to recipes or experimental procedures that do not necessarily produce positive results (or at least results that are consistent with the hypotheses of a particular experimental program). What may constitute a negative result in one research program could provide meaningful information to another, but such opportunities are hindered by the traditional publication scheme of primarily reporting positive results. To fill this gap, we propose the community should develop an Origins-of-Life equivalent to GenBank, where experimentalists from across the globe contribute soup recipes and the results of experiments to a publicly accessible database.

As an added advantage, such a community resource for sharing information on prebiotic soups could also house other information of broad interest, in particular methods for analyses of complex chemical mixtures. Untargeted analysis of complex chemical mixtures is notoriously challenging [13,124] so prebiotic researchers often fall back on methods optimized for the targeted analysis of biologically relevant compounds (e.g., proteinogenic amino acids, components of the citric cycle, or DNA/RNA components). However, to understand the emergence of many life-like phenomena, it will be necessary to track compounds that are not important in biology for which analytical methods are less readily available.

It is becoming easier to identify chemical formulae using extremely high-resolution mass spectrometry, and structures can also sometimes be inferred from mass-fragmentation spectra using searches in compound databases. However, the informatic pipelines are mainly designed for biological experimental data (e.g., metabolomics), which biases hits towards biological compounds [124]. Libraries developed for environmental analysis of complex mixtures (i.e., NIST (https://chemdata.nist.gov/) for pesticides, petroleum, and others) do include other small molecules, but are not as widely distributed. The recent development of computational methods for predicting the composition of prebiotic soups [125], may also be very useful in expanding our ability to identify unknowns in chemical mixtures.

Since prebiotic chemists all face similar analytical challenges, the prebiotic chemistry database that we are calling for should also include analytical pipelines and compound databases to help the community deal with the challenges of analyzing complex chemical mixtures. Furthermore, this resource could allow for dissemination of metrics and statistical tools for extracting useful insights from mixtures even when many of its component compounds cannot be identified. After all, as well illustrated by Van Krevelen diagrams [126], summary statistics can provide valuable insights into the overall characteristics of a chemical ensemble. Similarly, even when few if any mass spectral features can be tied to specific chemicals in a complex mixture, multivariate analyses (e.g., MDS, PCA) can be used to make comparisons across experimental treatments (e.g., Colón-Santos et al., 2019; Surman et al., 2019) [19,21] and to prioritize features for in-depth targeted analysis. A repository of relevant statistical methods and their corresponding scripts would therefore make a valuable third component of a community resource for prebiotic chemistry.

## 6. Conclusions: The Future of Messy Prebiotic Chemistry and Its Interplay with Reductionist Approaches

Although we have primarily focused our discussion on messy chemistry approaches to the study of the origin of life, we recognize that these approaches will always coexist with, and be complementary to, more reductionist research strategies for studying prebiotic chemistry [66,127,128,129]. Reductionist approaches allow chemists to home in on specific components of modern metabolism and explain how they might have originated in plausible prebiotic environments. For example, thanks to the work of organic chemists exploring the origins of genetic biopolymers, we know of several alternative polymer systems that could have preceded the appearance of RNA and would have been more easily synthesized and/or more stable under prebiotic conditions [130]. Similarly, research focused on components of the citric acid cycle has strengthened the idea that this or similar cycles could have been involved in prebiotic anabolic and catabolic processes [38,131,132,133,134,135]. However, reductionism cannot, by itself, tell us about the dynamic aspects of prebiotic chemistry or the appearance of emergent processes such as autocatalysis and adaptive evolution. As a result, systems chemistry approaches are needed to help us understand life as a general phenomenon without being biased by historically contingent features of life as we know it [136,137,138,139]. Only through bottom-up, untargeted methods can we determine what aspects of cellular biochemistry were inevitable for any living system given the specific chemistry of Earth, or were, instead, “frozen accidents” [140]. Likewise, because complex systems chemistry approaches can use soups and conditions that resemble other worlds it offers the potential to discover what other life-like phenomena might have emerged and what features they would be likely to manifest. Thus, more bottom-up chemical experimentation is needed, and this will depend on developing reasonably realistic and replicable prebiotic soups as inputs. In light of this we hope that the community of scientists studying the origins of life, regardless of their preferred experimental strategy, will pay more attention to the rational design of prebiotic soups and invest in a shared infrastructure for information sharing in prebiotic chemistry.

## Figures and Tables

**Figure 1 life-11-01221-f001:**
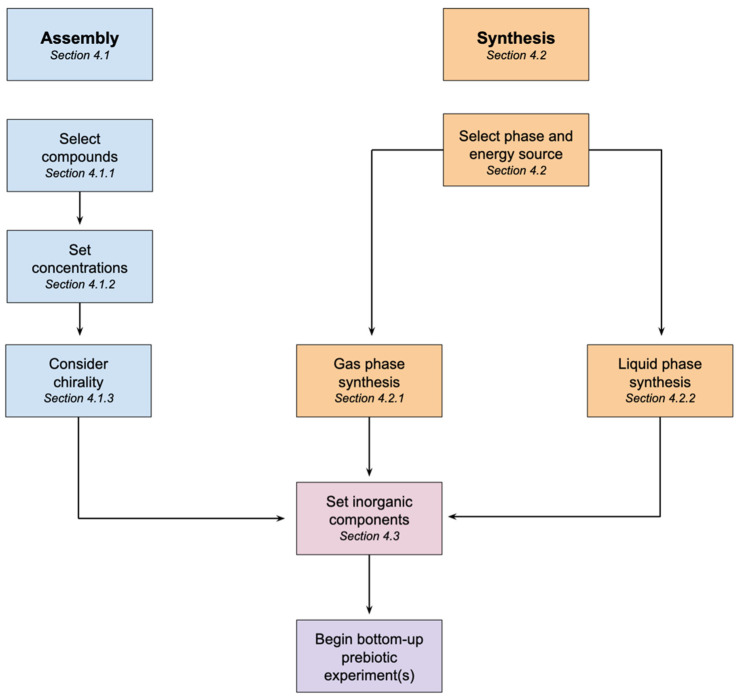
General workflow for assembling and synthesizing prebiotic mixtures.

**Table 1 life-11-01221-t001:** Comparison of approach to making prebiotic mixtures.

Consideration	Assembled Mixture	Synthesized Mixture	
		Atmospheric Synthesis	Liquid Synthesis
**Complexity of Products**	Low	High	High
**Procedure Difficulty**	Low	High	Low
**Control of Chirality**	High	Low	Low
**Control of Composition**	High	Low	Low
**Analytical Tractability**	High	Low	Low

## Data Availability

Not applicable.

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
