# Peer review of "The Prebiotic Kitchen: A Guide to Composing Prebiotic Soup Recipes to Test Origins of Life Hypotheses"

_life, 2021, doi:10.3390/life11111221_

Round 1

Reviewer 1 Report

Maurer et al. The Prebiotic Kitchen:….

The manuscript discusses an interesting question for the origin of life research field: How can one prepare (design or spontaneously form) the appropriate prebiotic soups?

As explained by the authors nicely, this question is relevant for historical reasons (what the soup on prebiotic Earth was) and bears practical implications (how one can prepare the most experimentally suitable soup). The paper provides the necessary background to probe these issues, although the description of certain aspects is sometimes left too genera. For example, the enormous issues associated with the applied concentration and the compounds’ chirality are discussed too briefly and with very few examples.

The paper is recommended for publication after the above-mentioned aspects are better developed and by extending the discussion in various directions:

  1. The formation of mixtures and the emergence of function (through the spontaneous formation of functional molecules) have been discussed many times in relation to Systems Chemistry. These aspects, and the appropriate references, can be added to the manuscript.
  2. The important phenomenon by which several compounds out of many were selected and prevailed to form the biological molecules we know today is called here ‘taming’. This has been studied in chemistry extensively, when research groups identified selection mechanisms that amplified molecules from the respective mixtures, including assembly into stable architectures and replication. Please elaborate.
  3. In my opinion, the conclusion and discussion section could/should include more data on currently studied systems. This could include the work of many groups around the world – some names that come to mind: Sutherland, Pascal, Hud….

Reviewer 2 Report

The manuscript by Vincent et al entitled “The Prebiotic Kitchen: A Guide to Composing Prebiotic Soup Recipes to Test Origins of Life Hypotheses,” discusses unconstrained chemical mixtures, and assemblies. It focuses on the technical challenges associated with designing, producing, and handling such mixtures. These mixtures have been used both historically and in contemporary research programs to explore prebiotic chemistry and chemical self-organization as it relates to the origin of living processes. The authors discuss how these mixtures have been produced, and characterized in the past, and they review the known sources of abiotic organics that would have been produced by or delivered to the early Earth, including quantitative delivery rates to the surface. The authors emphasize that constructing any chemical model of the early Earth will include uncertainty around exact chemical conditions and highlight the tension between chemical (and geological) realism and analytical tractability. They discussed important decisions to consider when preparing prebiotic soups, including the choice of input reagents, inorganics and concentrations, as well as recommendations for handling and storage of such mixtures. Finally the authors make a call to standardize approaches to prebiotic soups and for the establishment of an international database to record both the preparation procedures and the results of analysis for prebiotic soups.

After reading the manuscript over a couple times, I do not know what to make of it. My experience is in theoretical modelling of complex mixtures, and analysis of analytical data from complex mixtures, but not in bench chemistry itself. So perhaps the significance of the manuscript is lost on me, based on my lack of experience at the bench. The manuscript currently contains interesting ideas and useful pieces of information. But based on the way it is currently written, it's not clear what contribution it makes to the field. I think the current manuscript could be improved with revisions, but more significantly I think additional work in establishing the database and international standards would improve this manuscript and make a huge contribution to the field. I will try to explain my suggestions below.

The authors have included some useful summaries of the literature (particularly by reviewing the total organic input rates), and some useful tips for scientists working at the bench on prebiotic soups (like storing separate aliquots of samples to avoid freezing/thawing).  But the overall structure of the manuscript and the content seems disjointed. For example, why did the authors emphasize the two examples they included in the text? It is not clear from the current manuscript why these samples were prepared, or if they were analyzed. Were they just rhetorical tools to help explain the decision making process? Or were they prepared and analyzed, in which case, what were the results? From the included supplemental information it seems like these two examples were chosen to demonstrate how the authors would like to see the community report the design and preparation of the mixtures. But this was not made clear to the reader, and the role of the two examples included in the SI should be emphasized.

In reviewing the SI, the authors have clearly made decisions about best practice in reporting soup recipes, which include breaking the design goals and the recipe into distinct sections. But it's not clear why they made these decisions. It would be helpful if this approach was motivated and discussed in the manuscript. What other information is needed to make recipes reproducible from lab to lab? Do the authors recommend specific data formats for the recipes or the reagents? For example there can sometimes be ambiguities in reagent names, by using CAS numbers in addition to common names, the community might avoid potential confusion within recipes. I think specifically covering how soups should be documented could be covered in its own subsection within section 4 of the current manuscript

The abstract and introduction suggested the manuscript would be about principles of designing and handling prebiotic mixtures. But by the end of the manuscript I was not sure if the authors had proposed clear principles, or just suggestions. There seem to be 3 aspects of prebiotic soups the authors emphasized, 1) designing soups based on limited chemical knowledge of the early Earth, 2) preparing prebiotic soups, 3) handling and storing prebiotic soups. For each of these sections it would be useful to have a list (or table) of the principles that apply and perhaps an example of its application in the decision making process. This would clarify the exact principles the authors are proposing.

Also, the authors did not discuss analytical methods for characterizing complex mixtures. This seems to be a critical aspect of this approach and presents enormous technical challenges. I’m guessing the authors avoided this topic because it could be an entire research program (not to mention several manuscripts) in its own right. But if this manuscript is intended to help set standards and guide folks who are new to the field, it is important to mention the current analytical limitations and possible solutions.

Finally the authors called for a community effort in standardizing and recording unconstrained soups. I think the authors have clearly articulated the need for this within the community, and I completely agree. The space of possible experiments is far too vast for any one researcher or team to explore, a collective effort would provide significant advantages to the entire community. The authors have indicated what types of information would be recorded in the ideal case, and provided the examples in the SI, but they haven’t actually produced a framework that would be capable of effectively storing and sharing that information. I think the authors should make the first version of the example they proposed. Many different community wide repositories exist, including in fields that are closely related to the origins of life community. To what extent does the community need to build its own solutions, and how much can it piggyback on other communities and existing resources? It seems to me that such a repository is within reach, and the authors could produce the first version themselves. I think the important question here is how can the proposed prebiotic soup recipes, including the examples provided here- be stored digitally, so that they can be compared immediately to other recipes in terms of their design and production? If that can be accomplished the community can continue to use standard repositories for raw experimental data (such as https://www.ebi.ac.uk/metabolights/), and analysis code (such as github.com) and these could be linked back to the recipe.

By actually producing a repository of soup recipes the authors would provide not just a call to action but a specific point of focus for the community to begin the longer process of standardization across labs. Such an example would make the work in this manuscript significantly more impactful to the community.

Ultimately, I think the authors are working on an important topic - to understand the origin of organization in biology we must be able to characterize emergent complexity in unconstrained chemical mixtures. To begin that work we must be able to produce prebiotic mixtures reliably and reproducibly. I think the authors have provided some interesting insights here but the overall message of this manuscript is not yet clear. If the authors can go beyond a call to action, and create the first version of the tool they envisage, it would clarify what they are proposing and provide a huge resource to the field.

Reviewer 3 Report

Mixes ("soups") are discussed that could potentially play a role specifically for the planet Earth. Mixes are discussed on the assumption that life forms are emerging that are biochemically close to those currently known. This should be emphasized.

Apparently, the panspermia hypothesis is rejected by the authors. This should also be emphasized, since in the case of panspermia, the mixes in question may be other.

Although reductionism is criticized by the authors, in fact, the authors in one form or another use specific hypotheses of the origin of life on Earth. But so far no one knows how it was. Therefore, the proposed (knowledge) database on mixtures may not be complete.

Reviewer 4 Report

Dear Authors,

First of all, I would like to congratulate you for your didactic approach to the experiments in prebiotic chemistry. This articles deals with the necessity to explore the prebiotic chemistry with an appropriate methodology and the adequat approximations.

The article starts with a brief introduction, well documented but not exhaustive. The question and the postulate are well described. I suggest nevertheless to precise the term « robust », page 91. For you, what is a robust abiogenesis reaction ?

After this introduction, the general principles are proposed and the main strategies are described. I perfectly agree with these two realistic propositions. The structuration of this part is dense but clear.

The methodology is presented at item 4, line 246 and is well described. The main point of these paper is the realism of the approach for the methodology. For exemple, the choice to start experiment with racemic proportion is totally relevant and chemically reallistic.

I propose to add a discussion or precision at item 4,3, page 509. This part deals with the transport during the chemical reaction. All chemical reactions are occured out of the thermodynamic equilibrium. Why is important to take into account this for the prebiotic chemistry. Is it the fact to be out the equilibrium or is the nature of the entropic dissipation which is important ? I suggestion to improve this important part with a better theoritical introduction.

To conclude, I suggest to accept this paper. 

Best regards
